# The Development and Evaluation of a Novel Highly Selective PET Radiotracer for Targeting BET BD1

**DOI:** 10.3390/ph17101289

**Published:** 2024-09-27

**Authors:** Yanli Wang, Yongle Wang, Yulong Xu, Leyi Kang, Darcy Tocci, Changning Wang

**Affiliations:** Athinoula A. Martinos Center for Biomedical Imaging, Department of Radiology, Massachusetts General Hospital, Harvard Medical School, Charlestown, MA 02129, USA; ywang@mgh.harvard.edu (Y.W.); ywang151@mgh.harvard.edu (Y.W.); yulong.xu@mgh.harvard.edu (Y.X.); lkang5@mgh.harvard.edu (L.K.); darcytocci@brandeis.edu (D.T.)

**Keywords:** BET-BD1, bromodomains, positron emission tomography, radiotracer, imaging

## Abstract

**Background/Objectives:** Small molecules that interfere with the interaction between acetylated protein tails and the tandem bromodomains of BET (bromodomain and extra-terminal) family proteins are pivotal in modulating immune/inflammatory and neoplastic diseases. This study aimed to develop a novel PET imaging tracer, [^11^C]GSK023, that targets the N-terminal bromodomain (BD1) of BET family proteins with high selectivity and potency, thereby enriching the chemical probe toolbox for epigenetic imaging. **Methods:** [^11^C]GSK023, a radio-chemical probe, was designed and synthesized to specifically target the BET BD1. In vivo PET imaging evaluations were conducted on rodents, focusing on the tracer’s distribution and binding specificity in various tissues. Blocking studies were performed to confirm the probe’s selectivity and specificity. **Results:** The evaluations revealed that [^11^C]GSK023 demonstrated good uptake in peripheral organs with limited brain penetration. Further blocking studies confirmed the probe’s high binding specificity and selectivity for the BET BD1 protein, underscoring its potential utility in epigenetic imaging. **Conclusions:** The findings suggest that [^11^C]GSK023 is a promising PET probe for imaging the BET BD1 protein, offering the potential to deepen our understanding of the roles of BET bro-modomains in disease and their application in clinical settings to monitor disease progression and therapeutic responses.

## 1. Introduction

The intricate ballet of gene expression within the human body is orchestrated by a myriad of proteins, among which the bromodomain and extra-terminal (BET) proteins play a starring role [1]. Within this family, the BRD4 BD1 domain has garnered significant interest due to its critical function in recognizing acetylated lysine residues on histone tails, a process fundamental to the transcriptional machinery’s recruitment and the subsequent activation of gene transcription [2]. Such mechanisms are pivotal not just for the maintenance of regular cell cycle progression and proliferation but also for the modulation of inflammatory responses, situating BRD4 BD1 at the crossroads of epigenetics and disease pathology.

The importance of the BRD4 BD1 domain extends into the realm of disease, where its dysregulation has been linked to a spectrum of disorders, most notably cancer and inflammatory diseases [3]. In various cancers, including leukemia, breast, and prostate cancers, aberrant activity or overexpression of BRD4 has been observed to drive the transcription of oncogenes like MYC and BCL-2 [4]. Inhibiting the BRD4 BD1 domain has shown promising results in halting this transcriptional misregulation, resulting in the suppression of tumor growth and the induction of apoptosis in cancer cells [5]. Similarly, BRD4’s role in regulating pro-inflammatory genes positions it as a potential therapeutic target for inflammatory and autoimmune diseases, by potentially modulating the body’s inflammatory response [6].

With such a multifaceted role in health and disease, the BRD4 BD1 domain presents a prime target for therapeutic intervention. Preclinical studies have demonstrated the potential of small-molecule inhibitors targeting this domain, which have shown efficacy in displacing BRD4 from chromatin, leading to the altered gene expression profiles implicated in disease progression [7,8]. As shown in Figure 1, MS436, the first small-molecule nanomolar inhibitor targeting BD1, shows a selectivity tenfold greater for BRD4-BD1 over BRD4-BD2. This specificity is attributed to hydrogen bonds between MS436 and certain residues such as Pro82, Gln85, Lys91, and Asn140. In experiments with mouse macrophages, MS436 effectively reduced the transcriptional activity of BRD4, diminishing the production of pro-inflammatory cytokines like IL6 and nitric oxide [9]. Following MS436 [9], MS402 [10] emerged as an even more precise inhibitor, with nine times the specificity for BD1 over BD2, successfully reducing and alleviating colitis in mice by curbing the excessive growth of Th17 cells [10]. Another compound, Olinone, demonstrates a hundredfold greater selectivity for BD1 compared to BD2, enhancing the differentiation of mouse oligodendrocyte progenitors [11]. Such selective modulation of BDs suggests improvements in cellular regeneration, particularly beneficial for aging and neurodegenerative conditions [11]. GSK778 is an effective and selective inhibitor of the BET protein BD1 bromodomain, exhibiting a BET BD1 IC_50_ of less than 100 nM and a selectivity of 30- to 140-fold over BET BD2 [12]. GSK789, another highly selective BET-BD1 inhibitor, displays approximately 1000 times greater affinity for BD1 than BD2 [13]. Its efficacy parallels that of pan-BET inhibitors, yet it retains strong anti-inflammatory and immunomodulatory properties in vitro, underscoring its potential for clinical application [13].

Grasping the functional intricacies of the BD1 bromodomain is essential for unlocking the therapeutic possibilities of this target that is linked to various diseases, facilitating the creation of specialized BET inhibitors. While selective inhibitors of the BET domain have demonstrated potential in early trials, our comprehension of BD1 bromodomain’s precise roles in epigenetic control remains incomplete. Conventional in vitro methods, which often require the sacrifice of subjects, hinder the ability to conduct repeated studies on the same animals over time. Consequently, innovative approaches are indispensable for a deeper investigation into the distinct functions of each BD bromodomain. Positron emission tomography (PET) offers a sophisticated, non-destructive imaging solution that allows for the exploration of many complex questions within live subjects. By using domain-specific BET radioactive ligands, PET imaging can not only provide valuable information about the role of the BD1 bromodomain under physiological and pathological conditions but also assess the pharmacodynamics and pharmacokinetics of potential inhibitors by visualizing the engagement of the target, thereby determining the effective dose range for clinical benefits. Despite several domain-selective BET inhibitors entering clinical studies, the mechanisms of action are not fully understood, making PET imaging an urgent necessity for researching the mechanisms of BD bromodomains and discovering new domain-selective inhibitors. Until now, there have not been any reports of PET probes that specifically target the BD1 domain for preclinical applications. To tackle this issue, we are considering the creation of bromodomain-selective PET probes by incorporating positron-emitting isotopes, such as carbon-11 or fluorine-18, into well-established bromodomain-selective BET inhibitors [14].

GSK023 is a high-quality BET BD1 domain-selective chemical probe that exhibits 300–1000-fold selectivity for the BET BD1 domain (pIC_50_ of 7.8 against BRD4 BD1), with a phenotypic cellular fingerprint indicative of BET bromodomain inhibition [15]. Consequently, we have radiolabeled GSK023 by introducing a [^11^C]methyl group onto the amine of the pyridine ring, successfully synthesizing [^11^C]GSK023. Here, we describe the radiosynthesis and characterization of [^11^C]GSK023 as a potential selective BET BD1 bromodomain PET radioligand and demonstrate its pharmacokinetic properties in pilot imaging studies. 

## 2. Results

### 2.1. Optimizing GSK023 for PET Imaging: High Selectivity and Potency in BET BD1 Domain Targeting

In the selection of GSK023 as a PET radiotracer, several key characteristics stand out. Firstly, GSK023 demonstrates exceptional selectivity and potency for the BET BD1 domain, showing 300–1000-fold selectivity. This precision is crucial for accurate PET imaging to ensure specificity to targeted tissues or functions [16]. The tracer was identified and optimized through a meticulous structure-guided design process targeting a conserved Asp/His switch in the BET BD1 domain [17]. This process utilized advanced techniques such as X-ray crystallography, metadynamics simulations, and WaterMap analysis to fine-tune the interactions at the molecular level, ensuring effective binding and functionality within biological systems [15,18]. Moreover, GSK023’s structure facilitates the incorporation of carbon-11 (C-11), a radioactive isotope ideal for PET due to its short half-life of about 20 min, which permits rapid imaging and reduces radiation exposure [19]. The structural adaptability of GSK023 allows efficient tagging with C-11, enhancing the tracer’s utility in real-time tracking of biological processes with high resolution and sensitivity. These properties make GSK023 a highly effective PET tracer for studying the role of the BET BD1 domain in disease pathogenesis, offering insights with high specificity and accuracy in live imaging scenarios [15].

As demonstrated in Table 1, GSK023 exhibits exceptional selectivity and potency against the BET bromodomains, particularly BRD4 BD1, where it demonstrates very high inhibitory activity as reflected in its pIC_50_ and pKd values across different assay systems. For example, GSK023 maintains a fold selectivity of 1000 over BRD4 BD2 in the BROMOscan assay [20], indicating a very strong preference for the BD1 domain compared to BD2. Such high domain selectivity is crucial for a PET tracer, ensuring that it binds specifically to its target, which enhances the quality and reliability of imaging results [21]. Additionally, GSK023 shows remarkable specificity against a wide range of other potential targets. In the BROMOscan assay, it displayed a pKd of 316-fold selectivity over BRD4 BD1. This indicates that GSK023 has minimal off-target interactions, a crucial attribute for a PET tracer to avoid non-specific binding that can lead to ambiguous or misleading imaging data.

The combination of high potency, exceptional selectivity for the BRD4 BD1 domain, and minimal interactions with other proteins makes GSK023 an ideal candidate for development as a PET tracer. Its ability to distinctly target and visualize BET bromodomain interactions in vivo provides a powerful tool for studying the role of BET bromodomains in disease and for the development of targeted therapies [22].

### 2.2. Chemical Synthesis for GSK023 and [^11^C]GSK023

#### 2.2.1. Synthesis Summary of GSK023 and GSK023 Pre

The synthesis of GSK023 and its precursor involves several stages, as shown in Figure 1. Starting with the reaction of the initial materials, compound **1** (4.94 g, 35.00 mmol, 3.70 mL, 1.5 eq) and tert-butyl (S)-3-(aminomethyl)piperidine-1-carboxylate (5 g, 23.33 mmol, 1 eq), with K_2_CO_3_ (16.12 g, 116.66 mmol, 5 eq) in ACM (57 mL) at 80 °C for 5 h, resulting in the formation of compound **2** (7 g, 20.66 mmol, 88.56% yield, 99% purity). Next, compound **2** (1.5 g, 4.47 mmol, 1 eq) and 5-methyl-6-oxo-1,6-dihydropyridine-3-carbaldehyde (797.31 mg, 5.81 mmol, 1.3 eq) are treated with Na_2_S_2_O_4_ (2.34 g, 13.42 mmol, 2.92 mL, 3 eq) in a mixture of ethanol and water at 120 °C for 3 h, producing compound **3** (900 mg, 2.13 mmol, 47.63% yield). This intermediate (500 mg, 1.18 mmol, 1 eq) then undergoes acidification using hydrochloric acid in dioxane at 20 °C for 2 h, yielding compound **4** (460 mg, crude, HCl salt). Subsequently, compound 4 (500 mg, 1.39 mmol, 1 eq, HCl salt) is combined with 1-isopropylpiperidine-4-carboxylic acid (238.58 mg, 1.39 mmol, 1 eq), using HoBt (564.80 mg, 4.18 mmol, 3 eq), EDCl (400.64 mg, 2.09 mmol, 1.5 eq) in DMF (3 mL) at 20°C for 6 h, resulting in GSK023 Pre (103 mg, 93.59% purity by LCMS). In the final step, GSK023 Pre is methylated using methyl iodide (CH_3_I) and sodium hydride (NaH) in tetrahydrofuran (THF), beginning at 0 °C and continuing overnight to room temperature, to produce GSK023 with a yield of 56.1%, 54.9 mg.

#### 2.2.2. Radiolabeling of [^11^C]GSK023

The radiosynthesis of [^11^C]GSK023 was performed using a conventional methylation method, as shown in Figure 2. The synthesis began with the trapping of [^11^C]CH_3_I in anhydrous DCM (300 µL), to which the precursor GSK023 Pre (1.0 mg) and KOH (3.0 mg) were added. The reaction vessel was then heated to 100 °C for 3 min. The radioactive mixture containing [^11^C]GSK023 was quenched with HPLC mobile phase (0.5 mL) and purified via reverse-phase semi-preparative HPLC, resulting in a yield of 14.8–26.4% (decay-corrected from trapped [^11^C]CH_3_I). [^11^C]GSK023, identified by a retention time of 13.5 min, was diluted in water (15 mL) and processed through an SPE C-18 cartridge. After washing with water (10 mL), [^11^C]GSK023 was eluted using ethanol (1.5 mL), achieving a radiochemical purity of over 98%. Confirmation of [^11^C]GSK023’s identity was completed by co-injection alongside a non-radioactive GSK023. The final solution of [^11^C]GSK023 was then formulated in sterile saline (2.7 mL) and made ready for further in vivo experimentation [23].

### 2.3. In Vivo PET-CT Imaging with [^11^C]GSK023 in Rodents

The biodistribution and radiotracer uptake of [^11^C]GSK023 in target organs were investigated using a standardized uptake value (SUV), a parameter for quantifying the distribution of radiotracers in positron emission tomography (PET) studies. This metric calculates the radioactivity count rate in tissues against a theoretical count rate uniformly distributed throughout the body, adjusting for factors such as body weight, administered dose, and radioactive decay. 

Initially, the whole-body biodistribution of [^11^C]GSK023 in mice was studied. As shown in Figure 2, we selected five post-injection time points (5, 10, 15, 30, and 60 min) to assess the uptake, distribution, and clearance of the radioligand in key organs. [^11^C]GSK023 was highly distributed in blood-rich organs such as the heart, liver, and kidneys. In the heart and liver, the uptake of the radiotracer rapidly peaked post-injection and then gradually washed out. Conversely, in the liver and kidneys, the radioactivity levels rose quickly after injection, showed a slight decrease at about 10 min, and then gradually accumulated, suggesting metabolism pathways via the hepatobiliary and urinary systems.

To evaluate the specific binding of [^11^C]GSK023, blocking studies were conducted. Mice were pretreated with unlabeled GSK023 at doses of 4.0 mg/kg and 0.4 mg/kg 5 min before injection with the radiotracer. PET/CT imaging results and time–activity curves (TAC) for baseline and blocking studies are displayed in Figure 3. Notably, the uptake of [^11^C]GSK023 in the areas of interest significantly decreased in the self-blocking. The concentration of the radioligand in the target organs was reduced by approximately 45%, demonstrating the high specificity of the binding of [^11^C]GSK023.

Given the promising results in peripheral regions, we further investigated the biophysical properties of [^11^C]GSK023 in the brain (Figure 4). A whole-brain analysis revealed that 2 min post-injection, brain uptake reached a maximum SUV of 0.45 and then declined and stabilized around 0.2, indicating limited BBB penetration. The BBB permeability of the radioligand can be determined by key physicochemical properties such as molecular weight (M.Wt), lipophilicity (logP/logD), and total polar surface area (tPSA). Based on our experience, PET imaging probes with high BBB permeability typically have M.Wt < 500, a CLogP value around 4, and a tPSA value between 30 and 75 [14,19,22]. The molecular weight of [^11^C]GSK023 (487.7), its CLogP (3.2), and tPSA (59.5) might be the primary factors limiting its BBB penetration, even though these properties are close to but not entirely suitable for efficient BBB penetration. These characteristics might contribute to the observed limitations in BBB permeability (the M.Wt, CLogP, and tPSA of GSK023 were calculated using ChemDraw 14.0). Although the brain uptake level of [^11^C]GSK023 is limited, making it less suitable for imaging BET family proteins in the central nervous system (CNS), a significant reduction in radiotracer uptake was observed in the blocking studies. Based on the TACs, there was about a 50% reduction in whole-brain radioactivity in the self-blocking studies, further demonstrating the high specificity of [^11^C]GSK023 binding.

## 3. Discussion

In our study, the development of the PET radiotracer [^11^C]GSK023 as a highly selective probe for the BET BD1 family marks a significant advancement in epigenetic imaging. Its successful synthesis and subsequent in vivo evaluation via rodent PET imaging showed promising biodistribution characteristics. Notably, [^11^C]GSK023 exhibited significant uptake in peripheral organs (SUV = 1), while demonstrating limited uptake in the brain (SUV = 0.2), suggesting its potential utility for exploring the BET family’s protein expression in various diseases.

To contextualize the results of [^11^C]GSK023, a comparison with other PET tracers, such as [^11^C]Martinostat [24] and [^11^C]1a [25], reveals important distinctions. Unlike [^11^C]Martinostat, which provides good brain penetration and targets HDAC enzymes, [^11^C]GSK023’s limited brain uptake may restrict its applications in central nervous system disorders but proves beneficial for studying peripheral disease states where BET proteins are implicated, such as in certain cancers or inflammatory diseases. Additionally, further blocking studies with unlabeled GSK023 demonstrated a reduction of approximately 45–50% in the concentration of the radioligand within the target organs, further reinforcing its high specificity for BET BD1, paralleling the specificities seen with [^11^C]1a but with distinct peripheral organ distributions. These findings underscore [^11^C]GSK023’s utility in providing unique biodistribution data, essential for understanding the BET involvement in different physiological and pathological contexts.

## 4. Materials and Methods

Rodent PET/CT Acquisition. The Institutional Animal Care and Use Committee (IACUC), functioning under the Subcommittee on Research Animal Care (SRAC) at Massachusetts General Hospital (MGH), granted approval for all the experimental procedures detailed in this study. We utilized eight five-month-old male C57BL6 mice. For the study, we prepared a 1 mg/mL solution of GSK023 by dissolving it in a vehicle consisting of 10% DMSO, 10% Tween 80, and 80% saline. During the PET/CT imaging sessions, the mice were anesthetized with 1–1.5% isoflurane to ensure steady anesthesia levels throughout the scanning process. Each session involved positioning the mice in a Triumph PET/CT scanner (Gamma Medica, Northridge, CA, USA) and administering [^11^C]GSK023 intravenously in doses ranging between 8251 to 8547 KBq. Pre-administration treatments included a five-minute pretreatment with GSK023 at doses of 0.5 mg/kg and 2.0 mg/kg for one mouse each, or a vehicle for another mouse, all administered through a lateral tail vein catheter. Following the radiotracer injection, a dynamic PET scan lasting 60 min was performed, followed by a CT scan.

Rodent PET/CT Image Analysis. We reconstructed the PET data using the 3D-MLEM method, achieving a spatial resolution of 1 mm at full width at half maximum. The data were then refined with PMOD Technologies software (PMOD 4.01, PMOD Technologies Ltd., Zurich, Switzerland), based in Zurich, Switzerland, which processes the PET and CT images in DICOM format and aligns them with the brain atlas. We defined volumes of interest (VOIs) within the body and brain as elliptical regions, integrating CT structures with PET data for comprehensive analysis. Subsequently, we calculated time–activity curves (TACs) and derived standardized uptake values (SUVs), which represent the activity per unit volume after attenuation correction.

## 5. Conclusions

Overall, [^11^C]GSK023 represents a promising new addition to the arsenal of epigenetic imaging probes within PET studies, similar to [^11^C]Martinostat [24] and [^11^C]1a [25] but with distinctive characteristics that could potentially complement these existing tools. It serves as a valuable radioligand for investigating BET expression and aids in the in vivo evaluation of therapies targeting these epigenetic regulators. Considering the tracer’s limited brain uptake, our future direction includes optimizing [^11^C]GSK023 to enhance its blood–brain barrier penetration. Enhancing brain accessibility is expected to broaden its applications in neurological studies, thus providing a more holistic understanding of BET roles across a range of tissues. By delineating these specific characteristics and planning further research, we aim to substantiate the significance of [^11^C]GSK023 not only as a novel probe but also as a complementary tool in the evolving field of epigenetic PET imaging.

## Data Availability

The data supporting the findings of this study are included within the article and Appendix A.

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
