# Peer review of "The Development and Evaluation of a Novel Highly Selective PET Radiotracer for Targeting BET BD1"

_pharmaceuticals, 2024, doi:10.3390/ph17101289_

Round 1

Reviewer 1 Report

Comments and Suggestions for Authors

This study developed a novel PET 10 imaging tracer, targeting the BET family's N-terminal bromodomain (BD1) with high selectivity and potency, aiming to enrich the chemical probe toolbox. The evaluations showed that [11C]GSK023 has good uptake in peripheral organs and limited uptake in the brain.

The manuscript is well-written and supported by the results and discussion. However, some minor changes are required.

1. In the introduction, some relevant data should be highlighted to evaluate the better understanding for the readers.

2. In discussion, relevant comparisons must be added to better justify the results.

3. The conclusion should be better compiled separately and comprehensively.

4. The discussion is very short and discussion no relevant data.

Comments on the Quality of English Language

Some minor changes required

Author Response

Dear Reviewer,

Thank you for your letter concerning our manuscript entitled “Development and Evaluation of a Novel Highly Selective PET Radiotracer for Targeting BET BD1” (Manuscript ID: pharmaceuticals-3190613). Those comments are all valuable and very helpful for revising and improving our paper, as well as the important guiding significance to our research. We have studied comments carefully and have made correction which we hope to meet with approval. Revised portions are marked in red in the paper. The main corrections in the paper and the responds are as flowing:

  1. Response to comment: In the introduction, some relevant data should be highlighted to evaluate the better understanding for the readers.

Response: We have revised the introduction section to more effectively highlight key data. By presenting background information directly related to our research objectives more clearly, we provide a solid foundation to help readers understand the significance and context of our study. Additionally, we have updated the section on GSK023 to emphasize its high selectivity for the BET BD1 domain and added detailed data on GSK023's pIC50 value of 7.8 against BRD4 BD1, highlighted in red in the manuscript to offer a more comprehensive and substantiated discussion. Furthermore, we have consolidated all activity data, including IC50 values, into Figure 1 of the manuscript. Each data point in this figure is now accompanied by a detailed citation clearly indicating the source of these values. This enhancement ensures that readers can easily access and verify the information, enhancing the transparency and credibility of the article

  1. Response to comment: In discussion, relevant comparisons must be added to better justify the results.

Response: We have expanded the Discussion section to include detailed comparisons with other relevant studies, specifically with existing imaging agents [11C]Martinostat and [11C]1a. These comparisons not only link our findings to the existing body of knowledge, but also clearly show how our results are consistent with or different from previous studies, thus providing stronger support for our conclusions. These sections have been highlighted in red in the manuscript for easy identification.

  1. Response to comment: The conclusion should be better compiled separately and comprehensively.

Response:

The conclusion has been rewritten and is now presented as an independent section, highlighted in red within the manuscript. In the conclusion of our manuscript, we discuss [11C]GSK023 as a promising new PET imaging probe, highlighting its unique attributes that complement existing epigenetic imaging tools like [11C]Martinostat and [11C]1a. We emphasize its potential in investigating BET protein expression and outline future plans to optimize its blood-brain barrier penetration to enhance its applicability in neurological studies, thereby providing a more comprehensive understanding of BET roles in various tissues and strengthening its position as a valuable tool in epigenetic research.

  1. Response to comment: The discussion is very short and discussion no relevant data.

Response: Thank you for your comments regarding the brevity of our initial discussion. We have extensively expanded this section to include more detailed analyses and relevant data to bolster our points. The enhanced discussion now thoroughly details the development of [11C]GSK023, a PET tracer specifically selective for the BET BD1 protein, which is characterized by strong uptake in peripheral tissues and minimal brain uptake, making it particularly useful for studying peripheral diseases associated with BET proteins. We have also included a comparative analysis with other tracers like [11C]Martinostat and [11C]1a to highlight [11C]GSK023's distinct biodistribution and specificity. Further supported by blocking studies, these characteristics underscore its effectiveness and precision, establishing [11C]GSK023 as a crucial tool for investigating the role of BET proteins in various diseases. These revisions are clearly marked in red within the manuscript for easy identification.

We look forward to your further guidance and hope these revisions make our article more rigorous. We are committed to addressing all your concerns comprehensively.

Thank you again for your valuable comments.

Reviewer 2 Report

Comments and Suggestions for Authors

Dear Editor

I have read the manuscript with interest and attention. I think this work fits well with the topics and purposes of your journal.

The authors had an interesting idea on a topic of certain interest and very current for scientific research and they deal with it well.

Their results are excellent and this makes this work remarkable.

There are only a few small observations to make and a few points that the authors should correct.

Section 2.2 concerns the chemical synthesis of the molecules studied by the authors.

It is necessary to clearly indicate the purities of the starting reagents and of the compounds needed in the various steps followed for the overall synthesis.

The authors should indicate not only the overall yield, but also the individual stage yield would be very useful. Also, what by-products are formed? and how is purification carried out?

One part that is not sufficiently developed is the one concerning the chemical-physical characterization of the molecules, both of the final compound of interest and of the intermediate ones. I mean, there is no reason to doubt that the synthesis chain is the one indicated by the authors. However, I imagine that they will have done some analyses, to study the individual steps (NMR, IR and so on...) to prove that the products are exactly those indicated. What are the results of these analyses?

The authors state (based on the scientific literature?) that the molecule of interest, GSK023, "is a high-quality BET BD1 domain-selective chemical probe that exhibits 300−1000-fold selectivity for the BET BD1 domain, with a phenotypic cellular fingerprint indicative of BET bromodomain inhibition". Few more references here would be greatly appreciated for this and the other molecules too.

It should be useful to have a scheme that represents all the information reported about the activity, also reporting the IC50 values but most importantly citing from where these values come from. (this regards the part starting from line 58).

At page 5, the authors describe the radiolabeling of GSK023, according to Scheme 2.

Radiolabeling condition: [11C]CH3I, KOH, in 0.3 mL DCM, 3 min, 100℃, Radiochemical yield (RCY): 14.8–26.4%”. It should be depicted with the same format and design of the previous compounds.

A part from this, is this high selectivity also preserved in the resulting radiolabeled compound (in which a methyl group is added)? Are there any steric problems? And if so, do they affect the selectivity? The authors should comment

What is the final fate of the compound that the authors synthesize, that is, the molecule labeled with 11C? 11C decays quite rapidly to 11B, but the chemical properties of carbon and boron are quite different. So what happens to the radiolabeled molecule [11C]GSK023?

What can the authors say about the products? In particular about the aspects of biocompatibility and (cyto)toxicity?

This manuscript lacks a discussion of dosimetry for patient exposure to these labeled molecules. But this is probably beyond the scope and intention of this work. However, if the authors could add some comments and considerations in this regard, it would be greatly appreciated and would greatly improve the quality of their work.

Overall, however, this is a good manuscript and I wish to gently press the authors to make an effort to respond to these comments, so as to make their article publishable.

With my best regards.

Author Response

Dear Reviewer,

Thank you for your letter concerning our manuscript entitled “Development and Evaluation of a Novel Highly Selective PET Radiotracer for Targeting BET BD1” (Manuscript ID: pharmaceuticals-3190613). Those comments are all valuable and very helpful for revising and improving our paper, as well as the important guiding significance to our research. We have studied comments carefully and have made correction which we hope to meet with approval. Revised portions are marked in red in the paper. The main corrections in the paper and the responds are as flowing:

  1. Response to comment: Section 2.2 concerns the chemical synthesis of the molecules studied by the authors. It is necessary to clearly indicate the purities of the starting reagents and of the compounds needed in the various steps followed for the overall synthesis. The authors should indicate not only the overall yield, but also the individual stage yield would be very useful. Also, what by-products are formed? and how is purification carried out?

Response: We apologize for not clearly stating the purity of starting reagents and intermediates. We have revised the manuscript to include this information at all relevant steps. Additionally, we have reported the yields for each synthetic step and overall yield as you suggested. Regarding byproducts and purification methods, we have detailed the specific techniques used and the methods of impurity removal in the supplementary materials.

  1. Response to comment: One part that is not sufficiently developed is the one concerning the chemical-physical characterization of the molecules, both of the final compound of interest and of the intermediate ones. I mean, there is no reason to doubt that the synthesis chain is the one indicated by the authors. However, I imagine that they will have done some analyses, to study the individual steps (NMR, IR and so on...) to prove that the products are exactly those indicated. What are the results of these analyses?

Response: You rightly pointed out our lack of detailed physicochemical characterization. We have performed MS or NMR analyses at every step of the synthesis to verify the structure and purity of intermediates and final products. We have updated the manuscript with these analytical data, highlighted in red, and provided detailed descriptions in the supplementary materials, ensuring complete transparency of the synthetic route and compound identity.

  1. Response to comment: The authors state (based on the scientific literature?) that the molecule of interest, GSK023, "is a high-quality BET BD1 domain-selective chemical probe that exhibits 300−1000-fold selectivity for the BET BD1 domain, with a phenotypic cellular fingerprint indicative of BET bromodomain inhibition". Few more references here would be greatly appreciated for this and the other molecules too.

Response: We have updated the section on GSK023 reference supporting its high selectivity for the BET BD1 domain. Furthermore, we have added detailed data on GSK023's pIC50 of 7.8 against BRD4 BD1, highlighted in red in the manuscript to provide a more comprehensive and substantiated discussion.

  1. Response to comment: It should be useful to have a scheme that represents all the information reported about the activity, also reporting the IC50 values but most importantly citing from where these values come from. (This regards the part starting from line 58).

Response: we have incorporated all the activity data, including IC50 values, into Figure 1 of our manuscript. Each data point in this figure is now accompanied by appropriate citations that clearly reference the sources from which these values were obtained. This enhancement ensures that readers can easily access and verify the information, aligning with the recommendation to provide a comprehensive and verifiable overview starting from line 58. Thank you for guiding us to improve the clarity and credibility of our work.

  1. Response to comment: At page 5, the authors describe the radiolabeling of GSK023, according to Scheme 2. “Radiolabeling condition: [11C]CH3I, KOH, in 0.3 mL DCM, 3 min, 100℃, Radiochemical yield (RCY): 14.8–26.4%”. It should be depicted with the same format and design of the previous compounds. Apart from this, is this high selectivity also preserved in the resulting radiolabeled compound (in which a methyl group is added)? Are there any steric problems? And if so, do they affect the selectivity? The authors should comment

    Response: We have adjusted the description of the radiolabeling to maintain consistency with the formatting and design used for previous compounds, marked in red in the manuscript. Regarding the issue of methyl addition possibly affecting selectivity, as you pointed out, we have discussed in the manuscript whether the radiolabeled compound retains the high selectivity exhibited by GSK023. Adding the methyl group did not affect the spatial selectivity issue. We have conducted preliminary in vivo experiments to assess the selectivity of this radiolabeled compound and have included these results in the manuscript, marked in red.

  1. Response to comment: What is the final fate of the compound that the authors synthesize, that is, the molecule labeled with 11C? 11C decays quite rapidly to 11B, but the chemical properties of carbon and boron are quite different. So, what happens to the radiolabeled molecule [11C]GSK023?

Response: In PET imaging, molecules labeled with carbon-11, such as [11C]GSK023, are used for imaging via positron emission as carbon-11 decays to boron-11. While carbon and boron have significantly different chemical properties, the short half-life of carbon-11, approximately 20.4 minutes, and the typical timing of PET scans ensure that the decay of carbon-11 has minimal impact on imaging results during the process. After decay, the boron-containing molecule may further metabolize or be excreted from the body, but this typically does not affect the diagnostic efficacy of PET imaging.

  1. Response to comment: What can the authors say about the products? In particular about the aspects of biocompatibility and (cyto)toxicity?

    Response: The quantity of [11C]GSK023 used in PET imaging is extremely low, typically at nanomolar concentrations, insufficient to cause significant biocompatibility or cytotoxicity issues. Due to the minimal dosage and rapid decay, the radiotracer poses a low risk to biological systems, ensuring its primary use for safe and effective diagnostic imaging.

  1. Response to comment: This manuscript lacks a discussion of dosimetry for patient exposure to these labeled molecules. But this is probably beyond the scope and intention of this work. However, if the authors could add some comments and considerations in this regard, it would be greatly appreciated and would greatly improve the quality of their work.

Response: Thank you for suggesting the inclusion of dosimetry information. We recognize its importance but given the very small amount of [11C]GSK023 used and the short half-life of carbon-11, the radiation exposure to patients is considerably low. This detail supports the safety of [11C]GSK023 for clinical use, aligning with the focus of our manuscript. We believe this effectively addresses the fundamental safety concerns within the scope of our study.

We look forward to your further guidance and hope these revisions make our article more rigorous. We are committed to addressing all your concerns comprehensively.

Thank you again for your valuable comments.

Round 2

Reviewer 2 Report

Comments and Suggestions for Authors

Dear Editor

I have carefully read all the changes and edits made by the authors to their manuscript and I must say that I am completely satisfied with their responses.

I believe that this manuscript is now absolutely publishable.

With my best regards

Comments on the Quality of English Language

In my opinion, the English of this manuscript is more than fine.